# Team ÚFAL at CMCL 2022 Shared Task: Figuring out the correct recipe for predicting Eye-Tracking features using Pretrained Language Models

**Sunit Bhattacharya, Rishu Kumar** and **Ondřej Bojar**

Charles University

Faculty Of Mathematics and Physics

Insititute of Formal and Applied Linguistics

`bhattacharya,kumar,bojar@ufal.mff.cuni.cz`

## Abstract

Eye-Tracking data is a very useful source of information to study cognition and especially language comprehension in humans. In this paper, we describe our systems for the CMCL 2022 shared task on predicting eye-tracking information. We describe our experiments with pretrained models like BERT and XLM and the different ways in which we used those representations to predict four eye-tracking features. Along with analysing the effect of using two different kinds of pretrained multilingual language models and different ways of pooling the token-level representations, we also explore how contextual information affects the performance of the systems. Finally, we also explore if factors like augmenting linguistic information affect the predictions. Our submissions achieved an average MAE of 5.72 and ranked $5^{th}$ in the shared task. The average MAE showed further reduction to 5.25 in post task evaluation.

## 1 Introduction and Motivation

In the last decade that has seen rapid developments in AI research, the emergence of the Transformer architecture (Vaswani et al., 2017) marked a pivotal point in Natural Language Processing (NLP). Fine-tuning pretrained language models to work on various downstream tasks has become a dominant method of obtaining state-of-the-art performance in different areas. Their capability to capture linguistic knowledge and learn powerful contextual word embeddings (Liu et al., 2019) have made the transformer based models the work-horses in many NLP tasks. Pretrained models like the multilingual BERT (Devlin et al., 2019) and XLM (Conneau et al., 2020) have also shown state-of-the-art performance on cross-lingual understanding tasks (Wu and Dredze, 2019; Artetxe et al., 2019). In some cases like machine translation, there are even claims that deep learning systems reach translation qualities that are comparable to professional translators (Popel et al., 2020).

Language processing and its links with cognition is a very old research problem which has revealed how cognitive data (eg. gaze, fMRI) can be used to investigate human cognition. Attempts at using computational methods for such studies (Mitchell et al., 2008; Dehghani et al., 2017) have also shown encouraging results. However recently, there have been a number of works that have tried to incorporate human cognitive data collected during reading for improving the performance of NLP systems (Hollenstein et al., 2019). The CMCL 2022 Shared Task of multilingual and cross-lingual prediction of human reading behavior (Hollenstein et al., 2022) explores how eye-gaze attributes can be algorithmically predicted given reading data in multilingual settings.

Informed by the previous attempts at using pretrained multilingual language models to predict human reading behavior (Hollenstein et al., 2021) we experiment with multilingual BERT and XLM based models to test which fares better in this task. For the experiments with the pretrained models, we use the trained weights from Huggingface (Wolf et al., 2020) and perform the rest of our experiments using PyTorch[1]. Inspired by the psycholinguistic research on investigating context length during processing (Wochna and Juhasz, 2013), we experiment how different contexts affect model performance. Finally, we merged the principles of the "classical" approach of feature-based prediction with the pretrained-language model based prediction for further analysis. In the following sections, we present our results from a total of 48 different models.

## 2 Task Description

The CMCL 2022 Shared Task of Multilingual and Cross-lingual prediction of human reading behavior frames the task of predicting eye-gaze attributes associated with reading sentences as a regression

---

[1]https://pytorch.org/

task. The data for the task was comprised of eye movements corresponding to reading sentences in six languages (Chinese, Dutch, English, German, Hindi, Russian). The training data for the task contained 1703 sentences while the development set and test set contained 104 and 324 sentences respectively. The data was presented in a way such that for each word in a sentence there were four associated eye-tracking features in the form of the mean and standard deviation scores of the Total Reading Time (TRT) and First Fixation Duration (FFD). The features in the data were scaled in the range between 0 and 100 to facilitate evaluation via the mean absolute average (MAE).

## 3 Experiments

A total of 48 models of different configurations were trained with the data provided for the shared task. The different configurations used to construct the models are based on intuition and literature survey.

Thee models were primarily categorized as System-1 (sys1) and System-2 (sys2) models. For some word corresponding to a sentence in the dataset, System-1 models provided no additional context information. System-2 models on the other hand, contained the information of all the words in the sentence that preceded the current word, providing additional context. This setting was inspired by works (Khandelwal et al., 2018; Clark et al., 2019) on how context is used by language models.

All systems under the System-1/2 labels were further trained as a BERT (bert) based system or a XLM (xlm) based system. BERT embeddings were previously used by Choudhary et al. (2021) for the eye-tracking feature prediction task in CMCL 2021.

Corresponding to each such language models (bert and xlm), the impact of different fine-tuning strategies(Sun et al., 2019) on system performance was studied. Hence, for one setting, only the contextualized word representation (CWR) was utilized by freezing the model weights and putting a learnable regression layer on top of the model output layer (classifier). Alternatively, the models were fine-tuned with the regression layer on top of them (whole). This setting is similar to the one used by Li and Rudzicz (2021). However in our case, we experiment with a BERT and XLM pretrained model.

Additionally, we also performed experiments

with pooling strategies for the layer representations by either using the final hidden representation of the first sub-word encoding of the input (first) or aggregating the representations of all sub-words using mean-pooling (mean) or sum-pooling (sum). The rationale behind using different pooling strategies was to have a sentence-level representation of the input tokens. The impact of different pooling strategies has previously been studied (Shao et al., 2019; Lee et al., 2019) for different problems. In this paper, we analyze the effect of pooling feature-space embeddings in the context of eye-tracking feature prediction.

Finally, for the experiments where we augmented additional lexical features (augmented) to the neural features for regression, we used word length and word-frequency as the additional information following Vickers et al. (2021).

Constructing the experiments in this manner provided us with models with a diverse set of properties and in turn provided insights into how well the model behaves when all other things stay the same, and only one aspect of learning is changed.

## 4 Results

The results corresponding to the top 10 systems based on the experiments described above are shown in Table 1.

| Model | MAE |
|---|---|
| bert_sys2_augmented_sum_classifier | 5.251 |
| bert_sys2_unaugmented_first_classifier | 5.267 |
| bert_sys2_augmented_mean_classifier | 5.272 |
| bert_sys1_augmented_mean_classifier | 5.279 |
| bert_sys2_augmented_first_classifier | 5.295 |
| xlm_sys1_augmented_first_classifier | 5.341 |
| xlm_sys2_augmented_first_whole | 5.346 |
| bert_sys1_augmented_sum_classifier | 5.353 |
| bert_sys2_augmented_sum_whole | 5.367 |
| xlm_sys2_augmented_first_classifier | 5.373 |

Table 1: Top 10 best performing systems

It was observed that the maximum MAE scores (and the maximum variance of scores) for all the models was obtained for the attribute "TRT_Avg". The attribute wise variances corresponding to the test-data for all the models are shown in Table 2. Similarly, the mean values of the attributes for all models are shown in Table 3.

An analysis of the models based on the different experimental configurations are described in the

| FFD_Avg | FFD_Std | TRT_Avg | TRT_Std |
|---------|---------|---------|---------|
| 0.194   | 0.403   | 0.637   | 0.489   |

Table 2: Attribute wise variance of scores for all models

| FFD_Avg | FFD_Std | TRT_Avg | TRT_Std |
|---------|---------|---------|---------|
| 5.691   | 2.646   | 8.633   | 5.806   |

Table 3: Attribute wise mean of scores for all models

following sections.

## 4.1 System-1 vs System-2

Table 4 shows the average model performance across System-1 and System-2 configurations for both BERT and XLM based models (based on the average MAE values of the configurations). We see that for the BERT based models, the average MAE for System-1 is lower than that of System-2. But for XLM-based models, the difference is almost non-existent.

| Model     | Average MAE across models |
|-----------|---------------------------|
| Sys1_BERT | 5.66                      |
| Sys1_XLM  | 5.70                      |
| Sys2_BERT | 5.72                      |
| Sys2_XLM  | 5.69                      |

Table 4: System-1 vs System-2 performance across models

However, it should be noted that 12 out of the first 20 best performing models were System-2 models. Hence we posit that although the availability of the full sentence context is a factor for having more efficient systems, independently the factor does not seem to boost the overall performance much.

## 4.2 BERT vs XLM

Table 5 shows that there is only a tiny difference in average MAE for all four attributes (FFD_$\mu$, FFD_$\sigma$, TRT_$\mu$, TRT_$\sigma$) for all BERT vs XLM models . However, a brief look at Table 6 and Table 7 reveal that it was the XLM models that were responsible for slightly decreased MAE scores for 3 of the 4 attributes that were being predicted.

We also see that the amount of variance for XLM based models was also smaller for 3 of the 4 attributes.

| Model | Average MAE across models |
|-------|---------------------------|
| BERT  | 5.6920                    |
| XLM   | 5.6960                    |

Table 5: BERT vs XLM performance across models

| Model | FFD_$\mu$ | FFD_$\sigma$ | TRT_$\mu$ | TRT_$\sigma$ |
|-------|-----------|--------------|-----------|--------------|
| BERT  | 0.141     | 0.776        | 0.952     | 0.792        |
| XLM   | 0.236     | 0.045        | 0.349     | 0.204        |

Table 6: Attribute wise variance of scores for all BERT and XLM based models

| Model | FFD_$\mu$ | FFD_$\sigma$ | TRT_$\mu$ | TRT_$\sigma$ |
|-------|-----------|--------------|-----------|--------------|
| BERT  | 5.592     | 2.679        | 8.645     | 5.852        |
| XLM   | 5.789     | 2.612        | 8.622     | 5.760        |

Table 7: Attribute wise mean of scores for all BERT and XLM based models

## 4.3 Augmented vs Un-Augmented models

Fig. 1 shows that augmented models. i.e. models that were fed information like word-frequency and word-length along with the neural representation information before being fed to the regression layer performed better than models that used only contextual word embeddings resulting from pretrained language models. Table 8 and Table 9 show the 5 best performing models of this category sorted by their MAE.

| Model                                | MAE   |
|--------------------------------------|-------|
| bert_sys2_unaugmented_first_classifier | 5.267 |
| bert_sys2_unaugmented_mean_classifier  | 5.405 |
| xlm_sys1_unaugmented_mean_classifier   | 5.5   |
| xlm_sys2_unaugmented_mean_classifier   | 5.55  |
| xlm_sys1_unaugmented_mean_classifier   | 5.557 |

Table 8: Performance of 5 best Un-Augmented models.

| Model                               | MAE   |
|-------------------------------------|-------|
| bert_sys2_augmented_sum_classifier  | 5.251 |
| bert_sys2_augmented_mean_classifier | 5.272 |
| bert_sys1_augmented_mean_classifier | 5.279 |
| bert_sys2_augmented_first_classifier | 5.295 |
| xlm_sys1_augmented_first_classifier | 5.341 |

Table 9: Performance of 5 best Augmented models

The mean and variance of attributes across models of these families presented in Table 10 & 11 show that augmented models show way less vari-

| Model | FFD_$\mu$ | FFD_$\sigma$ | TRT_$\mu$ | TRT_$\sigma$ |
|---|---|---|---|---|
| Aug | 5.502 | 2.511 | 8.181 | 5.436 |
| Uaug | 5.88 | 2.78 | 9.086 | 6.176 |

Table 10: Attribute wise mean of scores for all Augmented and Un-augmented models

| Model | FFD_$\mu$ | FFD_$\sigma$ | TRT_$\mu$ | TRT_$\sigma$ |
|---|---|---|---|---|
| Aug | 0.017 | 0.004 | 0.015 | 0.007 |
| Uaug | 0.292 | 0.749 | 0.823 | 0.678 |

Table 11: Attribute wise variance of scores for all Augmented and Un-augmented models

ance in their predictions in comparison with neural-representation only model families.

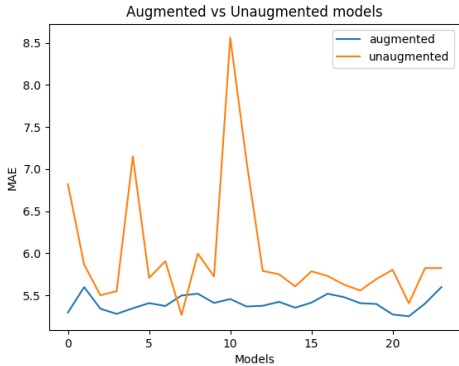

Figure 1: Augmented vs Un-augmented model performance. The x-axis represents the 24 different models of each category. The y-axis shows the MAE corresponding to each model.

### 4.4 Nature of representation of input tokens (Pooling strategies)

Fig. 2 shows that using the first sub-word token or the mean-pooled representation of the entire input gives lesser MAE scores than the sum-pooled representations. It was also observed that for System-2 family of models, the mean-pooled representations were associated with lesser MAE scores in comparison to the first sub-word representation. The attribute wise mean in Table 15 and attribute wise variance of model MAEs shown in Table 16 illustrates this point. Table 12, Table 13 and Table 14 show the 5 best performing models of this category sorted by their MAE.

### 4.5 Fine-tuning

Fine-tuning on large pretrained language models has become the standard way to conduct NLP re-

| Model | MAE |
|---|---|
| bert_sys2_unaugmented_first_classifier | 5.267 |
| bert_sys2_augmented_first_classifier | 5.295 |
| xlm_sys1_augmented_first_classifier | 5.341 |
| xlm_sys2_augmented_first_whole | 5.346 |
| xlm_sys2_augmented_first_classifier | 5.373 |

Table 12: Performance of 5 best first models

| Model | MAE |
|---|---|
| bert_sys2_augmented_mean_classifier | 5.272 |
| bert_sys1_augmented_mean_classifier | 5.279 |
| bert_sys2_augmented_mean_whole | 5.375 |
| bert_sys2_unaugmented_mean_classifier | 5.405 |
| xlm_sys1_augmented_mean_whole | 5.413 |

Table 13: Performance of 5 best Mean models

| Model | MAE |
|---|---|
| bert_sys2_augmented_sum_classifier | 5.251 |
| bert_sys1_augmented_sum_classifier | 5.353 |
| bert_sys2_augmented_sum_whole | 5.367 |
| bert_sys1_augmented_sum_whole | 5.402 |
| xlm_sys2_augmented_sum_classifier | 5.456 |

Table 14: Performance of 5 best Sum models

| Model | FFD_$\mu$ | FFD_$\sigma$ | TRT_$\mu$ | TRT_$\sigma$ |
|---|---|---|---|---|
| first | 5.549 | 2.505 | 8.434 | 5.615 |
| Mean | 5.57 | 2.538 | 8.416 | 5.636 |
| Sum | 5.954 | 2.894 | 9.05 | 6.167 |

Table 15: Attribute wise mean of scores for models with different input token representations

| Model | FFD_$\mu$ | FFD_$\sigma$ | TRT_$\mu$ | TRT_$\sigma$ |
|---|---|---|---|---|
| first | 0.036 | 0.004 | 0.118 | 0.054 |
| Mean | 0.047 | 0.005 | 0.118 | 0.048 |
| Sum | 0.383 | 1.082 | 1.374 | 1.139 |

Table 16: Attribute wise variance of scores for models with different input token representations

search after the widespread adoption of the transformer architecture. And unsurprisingly, our experiments reveal (Fig. 3) that fine-tuning of models give smaller MAE scores than training only the regression layers. The stark difference in the variance for the predicted attributes between fine-tuned models and regression only models (as illustrated in Table 17-18) further demonstrates the advantage of fine-tuning.

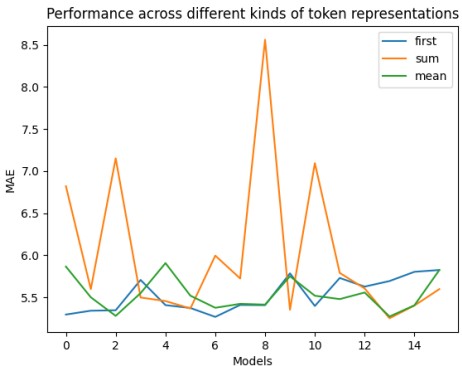

Figure 2: Model performance based on the nature of representation of input tokens.The x-axis represents the 16 different models of each category. The y-axis shows the MAE corresponding to each model.

| Model | FFD$\_\mu$ | FFD$\_\sigma$ | TRT$\_\mu$ | TRT$\_\sigma$ |
|-------|------|------|------|------|
| Aug | 5.502 | 2.511 | 8.181 | 5.436 |
| Uaug | 5.88 | 2.78 | 9.086 | 6.176 |

Table 17: Attribute wise variance of scores for fine-tuned models vs regression-layer only models

| Model | FFD$\_\mu$ | FFD$\_\sigma$ | TRT$\_\mu$ | TRT$\_\sigma$ |
|-------|------|------|------|------|
| Aug | 0.017 | 0.004 | 0.015 | 0.007 |
| Uaug | 0.292 | 0.749 | 0.823 | 0.678 |

Table 18: Attribute wise mean of scores for fine-tuned models vs regression-layer only models

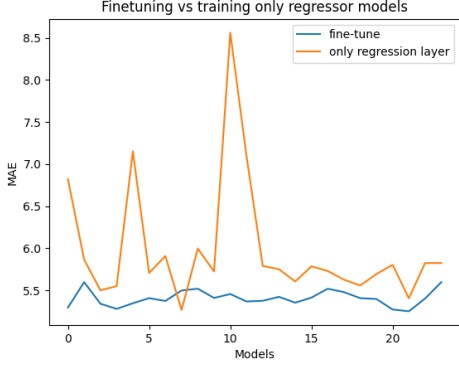

Figure 3: Fine-tuning vs training only regression layer in the models. The x-axis represents the 24 different models of each category. The y-axis shows the MAE corresponding to each model.

## 5   Conclusion

In this paper, we have described our experiments with different kinds of models that were trained on the data provided for this shared-task. We have identified five ways in which we can make better

systems to predict eye-tracking features based on eye-tracking data from a multilingual corpus. First, the experiments demonstrate that the inclusion of context (previous words occurring in the sentence) helps the models to predict eye-tracking attributes better. This reaffirms previous observations made with language models that more context is always helpful. Second, we find that XLM based models perform relatively better than the BERT based models. Third, our experiments show the advantages of augmenting additional linguistic features (word length and word frequency information in this case) to the contextual word representations to make better systems. This is in agreement with the findings from eye-tracking prediction tasks from last iterations of CMCL. Fourth, we see how different pooling methods applied on the input token representations affect the final performance of the systems. Finally, the experiments re-validate the approach of fine-tuning pretrained language models for specific tasks. Hence we conclude that contextualized word representations from language models pretrained with many different languages, if carefully augmented, engineered, and fine-tuned, can predict eye-tracking features quite successfully.

## 6   Acknowledgement

This work has been funded from the grant 19-26934X (NEUREM3) of the Czech Science Foundation.

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
