# OpenReview forum: "Team ÚFAL at CMCL 2022 Shared Task: Figuring out the correct recipe for predicting Eye-Tracking features using Pretrained Language Models"
_aclweb.org/ACL/2022/Workshop/CMCL_Shared_Task — CMCL Shared Task_

### Official Review · Reviewer_Yo9z · 2022-03-15
**Potentially informative contribution, but missing discussion of results**

**Rating:** 4
**Confidence:** 3

**Review:**


This paper presents the system description for a contribution to the CMCL 2022 Shared Task on Multilingual and Crosslingual Prediction of Human Reading Behavior. The authors propose to tackle to task using pretrained language models and performed a systematic study of the effects of model (mBERT vs. XLM), context information, input representation, lexical features, and fine-tuning.

# Pros

- The authors carried out a systematic study of a range of different model configurations for the task
- Different contributions to final performance are analyzed using an exhaustive grid-search

# Cons

- Figure 3, 4, 5: The x-axis label and ticks are not very informative. What are the different models? All possible model configurations? If yes, which configuration is which index? This information is crucial in order to allow the reader are more fine-grained interpretation of the results.

- The results should analyzed for statistical significance. This is especially important for cases with only marginal differences, as for example the comparison between BERT and XLM (Section 4.2): Here, the difference in MAE is probably not significant.

- The results are not discussed in relation to previous approaches for the task (especially regarding work on the challenge from 2021). In order to highlight the significance of the work, the authors should point out which results agree with previous findings, and which are novel.

- The performance of the best model is only marginally better than the mean baseline as reported in the challenge (MAE 5.72 vs. 5.73) and substantially worse than the mean baseline taking into account the target language (MAE 4.27). The authors do not discuss the performance of their models with respect to this baseline. It would be interesting to analyze why even the best of all 48 different models perform that poorly. Is it because of the high variance of feature values between the languages? Would these models perform better if they were trained and evaluated only on one language?

## Minor

### Abstract
- effects: affects
- lesser: smaller

### Introduction:
- Missing reference for “Huggingface”

### Results:
- Table 2 &3: Wouldn’t it be more infromative to report scores for the best performing model, instead of average over all tested models?
- The results presented in Figure 1 and 2 could be presented in Tables, as they show only 4 and 2 data points. This would save space and make the results more readable.

- Tables 6, 7, 8, and 10 are not at all mentioned in the text. If the results are not relevant, they can probably be removed?

---

### Official Review · Reviewer_Ck4k · 2022-03-20
**High potential, but further work is needed**

**Rating:** 5
**Confidence:** 4

**Review:**

The paper describes the system proposed for the CMCL2022 Shared Task on Multilingual and Crosslingual Prediction of Human Reading Behavior, and the different architectures implemented and compared by the Team ÚFAL.

**Pros**:
interesting approach, the different results of the various systems implemented and compared have a high potential in explaining both possible psycholinguistic insights and the computational models' natures.

**Cons**:
the paper is generally not precise enough. In particular: 1) no reason for the different architectures is provided, 2) the results are only reported, but not analyzed deep enough (i.e., the study lacks attempt in finding psycholinguistic-related explanations for the different performances)

**General suggestions**:
1) Give a justification for each different system (i.e., sys1, sys2, CWR, classifier, whole, first, mean, sum), that is, why you tried and compared these approaches.
2) collect the tables in an Appendix to have more space for results discussion


Minor

Introduction:
 - “have also shown state-of-the-art performance on cross-lingual understanding tasks”, insert at least one reference to a shared task or a study on cross-lingual understanding.
- reference to Huggingface

Experiments:
- “All systems under the System-1/2 label were further trained as a BERT (bert) based system or a XLM (xlm)”: labelS

Results:
- 4.1: “However, it should be noted than 12 out of the first 20 best performing models”: noted THAT

---

### Official Review · Program_Chairs · 2022-03-28

**Rating:** 6
**Confidence:** 5

**Review:**

The paper presents the system description of Team ÚFAL for the CMCL 2022 Shared Task on Multilingual and Crosslingual Prediction of Human Reading Behavior. The authors present a range of model comparisons for eye-tracking prediction. As pointed out by the reviewers, the paper has potential but requires some improvements. The descriptions should be more precise, especially regarding the motivation of the chosen model architectures and the discussion/analysis of the results.

We urge the authors to take the feedback from the reviewers into account and to improve their paper for the camera-ready deadline.

---

### Decision · Program_Chairs · 2022-03-28

Accept